# Silver Carp (*Hypophthalmichthys molitrix*) (Asian Silver Carp) Presence in Danube Delta and Romania—A Review with Data on Natural Reproduction

**DOI:** 10.3390/life12101582

**Published:** 2022-10-12

**Authors:** Abdulhusein Jawdhari, Dan Florin Mihăilescu, Sergiu Fendrihan, Valentin Jujea, Valeriu Stoilov-Linu, Bogdan-Mihai Negrea

**Affiliations:** 1Department of Anatomy, Animal Physiology and Biophysics, Faculty of Biology, University of Bucharest, 91-95 Splaiul Independenței Str., 050095 Bucharest, Romania; 2Department of Animal Production Techniques, Almussib Technical Instated, Al-Furat Al-Awsat Technical University, Babylon-Najaf Str., Najaf 54003, Iraq; 3Non-Governmental Research Organization Biologic, 14 Schitului Str., 032044 Bucharest, Romania; 4CE-MONT, Mountain Economy Center of the “Costin C. Kiritescu”, National Institute of Economic Research—INCE, Romanian Academy, 49 Petreni Str., 725700 Vatra Dornei, Romania; 5Geography Department, Geography and Geology Faculty, Doctoral School of Geosciences, “Alexandru Ioan Cuza” University of Iasi, 20A Carol I Str., 700505 Iasi, Romania; 6Doctoral School of Applied Sciences (Biology), Ovidius University of Constanta, 58 Ion Voda Str., 900525 Constanta, Romania

**Keywords:** *Hypophthalmichthys molitrix*, Danube Delta, Asian carp, silver carp, invasive species, reproduction, global warming

## Abstract

The Danube River has a large hydrographical basin, being the second largest river in Europe. The main channel flows through seven European countries with many species of fish inhabiting it. In this review we focused on the invasive species silver carp (*Hypophthalmichthys molitrix*), which plays an important ecological and economic role in its original habitat, but since introduced in Europe’s rivers, the species has posed a serious ecological risk under global warming. In this review paper, we gathered data regarding silver carp, such as when and how it entered the Danube Delta and the water temperature suitable for its growth and reproduction, mainly in the context of global warming, as well as the nature of nutrition and the ecological risk the species poses.

## 1. Introduction

As the second-largest river in Europe, the Danube has a big hydrographical basin measuring roughly 817,000 km^2^. The main channel crosses seven nations, before reaching the Black Sea and a large part of its lower sector flows into Romanian territory, where it creates the Danube Delta. New studies or monitoring programs led by international organizations or associations such as the International Commission for the Protection of the Danube River (ICPDR) and the International Association for Danube Research highlight the importance of the Danube for nature and biodiversity. A potential “invasion gateway” in Europe, the Danube River is a part of the Southern invasion corridor [1]. The intricate Danube–Black Sea agroecosystem is a rich “ichthyosystem”, formed by a rare combination of integrated biotopes and biocoenoses linked to forces and counterforces in time and space. The Danube Delta’s structural and functional bonds in the connectivity of the Danube and the Black Sea are revealed by the fish species captured there. In addition, the fact that 57.26% of the fish species found in the Lower Danube, Danube Delta, and northwest Black Sea use two or three of the subsystem habitats, demonstrates the delta’s crucial importance. Therefore, this convergence region can be thought of as a dynamic “ichthyosystem” that has demonstrated a high degree of adaptability, resilience, and flexibility over geological time but has become much more sensitive to environmental perturbations as a result of human impact over the past century as the introduced non-native fish species have a large impact on this system [2].

The term Asian carp collectively refers to a group of four species that are also now found in the Danube River; they are: the grass carp (*Ctenopharyngodon idella*, Valenciennes 1844), bighead carp (*Hypophthalmichthys nobilis*, J. Richardson 1845), black carp (*Mylopharyngodon piceus*, J. Richardson 1846), and silver carp (*Hypophthalmichthys molitrix*, Valenciennes 1844) [3].

Asian carp are widely cultured and traded all over the world due to their fast growth rate, easy cultivation, high feed efficiency ratio, and high nutritional value [3,4,5]. Silver carp, *Hypophthalmichthys molitrix* (Valenciennes 1844) is one of the most commonly raised freshwater fish species throughout the world due to its wide availability, low cost of aquaculture production, high feed efficiency ratio, and nutritional value [6]. It is well known to be rich in proteins, polyunsaturated fatty acids, lipid-soluble vitamins, and micronutrients [7,8]. Asian carp of the genus *Hypophthalmichthys* are valued as food resource not only in eastern and southern Asia, where they are considered a common aquaculture species, but also in global aquaculture production [9]. As a planktivorous fish species, *Hypophthalmichthys* spp. has biological traits that make it both ideal an aquaculture species and effective invaders (traits such as quick growth and reproduction, wide environmental tolerance, and general eating preferences); thus, in many of the territories where they have been introduced, Asian carp are now regarded as invasive species in areas outside their native range, especially in the USA [10].

*Hypophthalmichthys molitrix* (Valenciennes 1844) and *Hypophthalmichthys nobilis* (J. Richardson 1845), known as silver carp and bighead carp, respectively, play an important economic role in freshwater fish farming. Looking through fish harvest reports is adequate to comprehend the significance of these species. For instance, silver carp ranked second in the world’s aquaculture in 2018 with a global production of 4,822,794 tons. Iran, Bangladesh, China, and India are the top producers and exporters [11]. A total of 1.369 tons of bighead carp were collected in 2019 in Hungary as part of the so-called “carp polyculture system,” accounting for around 7 to 10 percent of the nation’s overall fish production [9]. Additionally, the silver carp has been introduced to about 80 other countries and still serves a significant ecological function in its native habitat (East Asia), successfully reproducing in 23 countries [12].

The ecological impact of Asian carp in North America is already well known [13]. Optimal temperature and water flow are the two primary parameters involving their reproduction. While the two species of Asian carp need a minimum temperature of 18 °C to breed [14], this is not a limiting factor in the majority of Europe. While silver carp females are less demanding in this sense, bighead carp females need strong water currents in the spawning areas for egg development to take place [15].

In Europe, the introduction of these species began in the 1960s; however, in Western Europe, the potential of wild reproduction was not taken into account [16]. The first proven reproduction of bighead carp was reported in the Po River and its associated canal system [17]. The use of the two species (and their hybrids) in aquaculture production is a crucial component of the so-called “carp polyculture system” under pond farming conditions, which is a prevalent practice in Central and Eastern Europe, further complicates the situation in Europe [18].

The Danube and Tisza River basins in this region provide the environmental conditions needed for spawning. There have been reports on the two species spawning observations in the Danube’s Romanian and Serbian parts [19]. Asian carp effects on the ecosystem are still largely unknown in Europe, where the species has also been introduced. Some sources claim that the species is established at least in the Danube River, e.g., [20], but others report they are only present in Europe through stocking and escapes [13]. In the rivers of Western Europe, Asian carp were introduced at the beginning of the 1960s and are typically thought to be incapable of reproducing. However, there have been no studies on the stability of their population or the effects they have on the ecosystem. Bighead carp invasions are likely underreported more frequently than silver carp ones, which are frequently found together with bighead carp and have a propensity to jump out of the water when boats pass nearby it [21], thus being easily observable. This work aims at highlighting the Asian carp as an invasive species, with an emphasis on *Hypophthalmichthys molitrix*. The invasiveness rate is facilitated by the current climate change context, as higher temperatures lead to the better reproduction of the species in question, which involves yet unknown long-term consequences for the Danube Delta. 

## 2. External Morphology and Biology

The silver carp’s overall form is described as deep-bodied and moderately compressed laterally [22]. The number of vertebrae is 37 from the head to the anus. The area of the gill spines transformed into a filter apparatus, by means of a conjunctive-cartilaginous tissue with holes that retain particles of approximately 8–10 µm. It has a weakly pronounced variability.

Sexual dimorphism is weakly highlighted: males have a smaller waist, and on the first radius of the pectorals, when palpated, there are asperities. The common size is 45–55 cm, with a maximum of 105 cm and 12 kg. The fins do not have any spines, and the scales are exceedingly tiny and cycloid. It is dark gray above and off-white below, with irregularly shaped and placed splotches of dark gray to black all over the body. When the fish reaches the age of about 8 weeks, this pattern starts to emerge. Silver carp also have unusually big heads and mouths [22] (Figure 1) and are often mistaken with bighead carp.

The terminal mouth is not expandable, and the projecting mandible and premaxilla together produce stiff, bony lips. The eyes have a distinct ventral orientation and are situated anteriorly on the head. Between the pelvic fins and the caudal fin base lies a smooth keel [22,23] (Figure 1).

It prefers slow-flowing and stagnant fresh waters, with a broad sheen and devoid of vegetation, rich in microalgae and organic suspensions, which it feeds on by filtration. In warm seasons it spends most of its time filtering, when it moves slowly in the water mass or near the surface, in larger or smaller groups, usually constituted by age. When it encounters obstacles (for example, fishing nets or dams) it tries to jump over them. If in the native area, sexual maturity is reached at the age of 3–4 years; here, in Eastern Europe it is 1–2 years later, at a body weight of 3–4 kg [22].

In its native areas (East Asia), reproduction takes place at water temperatures between 18–22 °C in rivers, during periods of flooding, in eddy areas. Reproduction takes place in groups of 15–25 individuals. Spawns are pelagic. Hatching occurs after approximately 24 h, and the brood spreads in the overflow areas of the rivers.

In the fish farms of the Danube Delta area, reproduction can only be achieved artificially by hormonal stimulation with pituitary suspension. The best results were obtained in the first half of June, when the water temperature is between 22 and 25 °C During feeding it retains particles from 8 to 100 µm, mostly consisting of planktonic algae of all categories, including blue, zooplankton, and organic detritus. There are some opinions that the species is resistant to the toxins of some species of blue algae, and others that it has the ability to detect and avoid fields with such toxic algae. The rate of growth mentioned in Danube Delta ponds area is below that of carp in the first summer (approx. 10–20 g), but after this period it is superior, having a very high growth capacity in conditions of sufficient food so at the age of 5 years it can reach a weight of 7–8 kg and at 6 years can exceed 10 kg [22].

## 3. When and How the Species Entered the Danube Delta and Romania

Due to the abundance of nutrients that the Danube is carrying out, fish move to these areas most likely to take advantage of the water’s distinctive richness. Numerous native freshwater fish species regularly migrate into these zones in search of food. Freshwater fish species typically migrate at specific seasons. In order to spawn in the most naturally protected regions, fish migrate in the spring. In the last decade, in the Danube Delta area it has been proven that specimens escaped the environment and reproduced in natural waters [22]. The species has “escaped” from the aquaculture systems many times through the water supply systems not equipped with sieves, and thus the fry entered the natural waters (the hydrographic network of Romania is shown in Appendix A). The seasonal migration of freshwater fish is very closely related to the water temperature [24]. Other detected assessments of fish data for the entire Danube River showed that the non-native *Hypophthalmichthys* molitrix was among the 20 most abundant species, ranking seventh and eighth, respectively [25].

The adaptation of Asian carp in Romania (Figure 2) begins in 1957 with the importation of 100 young Asian grass carp from the former USSR. The introduction of younglings of the other species of Asian carp (also silver carp) from the People’s Republic of China in the early 1960s continued this action. These species were initially introduced in a few fish farming operations and research centers in western and central Romania, as well as a few farms in the Danube Delta [26,27]. The young Asian carp likely strayed into the wild accidentally after escaping from certain fish farms, perhaps in the Danube River itself or one of its major tributaries. The exact dates of the incident and the number of specimens which escaped into the wild are unknown. The significant floods that occurred in the middle of the 1970s may be a factor in some of these accidents.

Since then, more and more Asian carp have begun to be caught in the Danube Delta and in the upper course of the Romanian river sector of the Danube until the end of the 1970s. In the middle of the 1990s, their weight as a percentage of the annual total capture reached more than 100 tons. Spread of invasive alien species (IAS) to the Bulgarian Danube River section may be additionally facilitated by international shipping, boating, aquaculture, and fishing activities; for example, it was considered that it was intentionally introduced along with the herbivorous Asian carp *Aristichthys nobilis* (Richardson 1845) to Mechka fishponds or other fish farms along the Danube River located upstream [26,28].

The Asian silver carp is the most productive and economical species for directed growth (fish culture) due to the fact that it has a better growth rate than carp and does not need to be fed, consuming the phytoplankton mass. Since 1981, it has also started to be recorded in catches made in natural waters of the Danube Delta, especially in the Razim-Sinoie complex, the Old Danube, the Danube upstream of Isaccea, and some large lakes. In the Danube, the catches are usually made in the spring, with nets, on the occasion of pontic shad fishing, and in the other areas, in the fall, by nets. Although the productions made from the natural environment are transferred to the “Fito” (grass feeding Asian carp) category, including here the three species of East Asian cyprinids, it can be stated that *Hypophthalmichthys molitrix* is the dominant species so that at least 80% of the quantities reported in the Danube Delta Biosphere Reserve area belong to it [22].

In the study on fish fauna from lakes of the fluvial Danube Delta [29], the presence of fish communities that vary among lakes can be considered a good indicator of the ecological state of lakes, using fish catches from the Gorgova-Uzlina lake complex, it was among the dominant species of carp, among the 37 species captured.

The study of ichthyofauna realized by Năvodaru et al. [30] presents the situation of the fish communities from the Rosu-Puiu lake complex, part of the “marine delta area” of the Danube Delta. In this lake complex, there were analyzed 29 species of fish (5 exotic species and 24 native species, including Asian silver carp). Another complex study about the Cyprinidae health condition in Romania, among them the Asian silver carp, indicates that the knowledge of fish illnesses is of great importance, as long as diseases represent restrictive factors of fish production [31].

A rather technical form on how to establish the taxonomic relationships within the Cyprinidae family was required because it is one of the largest families of fish in the world and a well-known component of the East Asian freshwater fish fauna. The majority of research on silver carp has used single mitochondrial DNA genes to assess population or low-level taxonomic relationships [32].

Among the fish species that currently show an obvious biological progression and hold the largest share in the captures of the Prut River basin (within the borders of the Republic of Moldova) is the *Hypophthalmichthys molitrix*, in the lower sector, along with many species studied in the research on the ichthyofauna of the Prut river [33].

The presence of the species *Hypophthalmichthys molitrix* is also observed in the Tansa-Belcesti lakes, situated in the northeastern part of Romania, on the Bahlui River course, where a numerical and gravimetric inventory is made to be able to observe the dominance of higher economic importance species but also the elements of parasitofauna [34].

Năstase et al. [35] placed on the map of Romania the presence of silver carp within the river basin of the Mures river, through the investigations regarding ichthyofauna from four localities sites of lower Romanian Mureș River. Regarding origin most of the species are native, but exotic species like *Hypophthalmichthys molitrix* appear as new species in the area caused by accelerated eutrophication, the results of intensive human activity in sense of increased pollution in the condition of continuous climate change.

In the Prut River, in localities such as Giurgiulesti or Caslita, an active entrance from the Danube River and an increase in the number of Mediterranean species was established, and, from the Cyprinidae family, the biggest share is held by *Hypophthalmichthys molitrix* [36].

A number of 21 species were recorded, including *Hypohthalmichthys molitrix*, starting from the premise that fish from the Timiș river represent a biological indicator in assessing the water quality of this river [37], an aspect that was treated relatively similarly in the work on fish fauna from the lowland Mureș River (Romania) by Teclean et al. [38]. Among the farmed freshwater fishes, *Hypophthalmichthys molitrix* has attracted great attention due to its increasing production in the European Union, where the silver carp is produced most notably in Hungary and Romania [39]; therefore, the presence in our country is significant for this species. Additionally, Luca et al., [40] used the vertebrate mitochondrial genome as an important model system for studying molecular evolution, organismal phylogeny, and genome structure in order to establish the phylogenetic relationships between analyzed species from the Cyprinidae family in Romania.

In 2014, a study was carried out on the species of Stanca Costesti Lake, located on the border between Romania and the Republic of Moldova, where the results of a fish fauna study in the area were based on 29 species, with important resonances being *Hypophthalmichthys molitrix* [41].

In the branches of the Danube Delta and in the adjacent flowing waters of the Danube Delta Biosphere Reserve (RBDD), 66 species of fish have been recorded, of which two-thirds have commercial value. The fish species are mostly native, while only six are exotic species, including *Hypophthalmichthys molitrix* [42].

Other studies present a new location, where there were described 43 fish species in Razim Lake, with 39 native and 4 non-natives (including *Hypophthalmichthys molitrix*). The aim was to determine the ecological status of the fish fauna of Razim Lake under the conditions of the water salinity changing [43].

In 2009, it was identified that the number of species that usually live in the lowland waters is increased in the lower Mureș River (29 species) and is decreasing in the Someş and Crişuri Rivers system (22 species), where the riversides are embanked. The majority of the fish species that live in the lowland rivers use the secondary branches and adjacent ponds, and canals as natural refuges as optimal biotopes for their spawning. Due to the entity of the adjacent channels, the opportunistic *Hypophthalmichthys nobilis* (J. Richardson 1845) and *Hypophthalmichthys molitrix* are increasing in number in the lower Mures [44,45] and have identified the fact that the Lower Siret Valley is very rich in ichthyofaunal diversity, including *Hypophthalmichthys molitrix*. At the same time, a few specimens the species were found in the Putna-Vrancea Natural Park and the Dumbravița fisheries complex Ramsar sites.

A study aiming to monitor fish parasites in Romania, as well as their importance for pathology and public health, monitored more than 50 wild fish species, including *Hypophthalmichthys molitrix* [46].

As a mention of the distribution of silver carp in the territory of Romania, (Figure 3), there was also the monitoring of the Someș River, in 2013, by investigating the qualitative and quantitative changes of fish assemblages in the polluted areas [47].

## 4. Water Temperature Suitable for Reproduction of *Hypophthalmichthys molitrix*

Ecosystems supporting fisheries are negatively impacted by climate change and fluctuation. Freshwater ecosystems may experience widespread effects from climate change due to changes in biodiversity patterns, species abundance and distribution, biological interactions, phenology, and the physiology, performance, and fitness of animals. Romania is included in the region of Central and Southern Europe which is regarded as one of the most vulnerable to climatic changes [48]. Thus, as a result, climate change affects the water level, temperature, and ambient thermal regimes of the Danube River and the variety and richness of the ichthyofauna in the lower sector of the river [49].

Romania’s summer temperatures over the past 10 years have been the highest in the country throughout the previous century [50,51]; additionally, catastrophic climate events such as extreme precipitation or drought are linked to global warming and can alter the water cycle, leading to anomalous oscillations in water level [52]. In the past 3 years, studies have been conducted to assess the effects of the Danube River’s water temperature and level regime on the structure and variety of the fish stock (between 2016 and 2019 in Braila station) [53].

In order to demonstrate the climate warming and the thermal trend of the Danube River and Danube Delta in the May–June period, we undertook our own climate demonstration based on raster data extracted with the help of the WorldClim source. This time period is when *Hypophthalmichthys molitrix* is natural breeding if environmental conditions are favorable and the waters are warm enough.

The raster data were taken from the WorldClim website, where monthly climate data starting in 1968 are stored [16,54,55]. The data stored on the site are provided by the Climatic Research Unit, University of East Anglia, using WorldClim 2.1 for bias correction. The variables available are average minimum temperature (°C), average maximum temperature (°C), and total precipitation (mm). The spatial resolution is 2.5 min (~21 km^2^). Each download is a “zip” file containing 120 GeoTiff (.tif) files, for each month of the year (January is 1; December is 12), for a 10-year period. The data were processed in QantumGis (open source). Clip raster by mask layer was made to process the data of the research area (Danube and Danube delta), raster to polygon (vector) conversions for data extraction.

Because we were missing data from the CRU source, WorldClim 2.1 for the years 2018–2019, we completed with data from the stations measuring the parameters of the Danube waters in Tulcea, Galați, and Brăila locations for these years, and the water temperature during June was expressed as follows:

The mean annual highest water temperature was recorded in 2019 (17.16 to 8.39 °C), and the lowest was recorded in 2017 (16.25 to 9.07 °C) (Figure 4). Galati station recorded its mean annual maximum water temperature in 2019 (16.86 to 8.62 °C) and its mean annual minimum temperature in 2017 (16.35 to 9.41 °C) (Figure 4). In Tulcea station, the mean annual water temperature ranged from 16.20 to 9.28 °C in 2017 to 17.15 at 8.52 °C in 2019. (Figure 4), [14,54,56].

The conclusions regarding minimum temperature, measured in the middle of May, on CRU temperature data, registered an increase of 1.56 °C in the period 2010–2021. In the same time maximum temperature, measured in the middle of June, registers an increase of 1.26 °C in the period 2010–2021.

In Figure 5 it can be seen that the high temperatures are located in the upper and middle course until the entrance of the Danube in the Danube Delta area. There are only small points with slightly lower temperatures in 2010, so in 2021 the high temperatures of water in May–June occupy the entire course of the river, including in the Danube Delta.

The temperature is increasing on average of 0.11 °C per year, predicting an increase by at least 1.14 °C in the next decade. There is a possibility that this increase will be an accelerated one because in the period followed by the study, an acceleration of this increase towards the year 2021 was observed. The minimum temperature is maintained, with small fluctuations around 25 °C, registering an important increase in 2021 (27.24 °C). The maximum temperature increases visibly, registering a significant evolution from 29.64 °C in 2010 to 30.9 °C in 2021. A significant increase in values is observed between 2010 and 2021, namely 1.26 °C.

The temperature in the presented graphs shows its evolution for the last half of May and the first half of June, which coincides with the breeding period of Asian silver carp.

The annual mean June temperature is likely to be at least 1 °C warmer than pre-industrial levels, and it is equally likely to be at least 1.5 °C over the next 5 years.

In conclusion, the phenomenon of increasing temperatures for May–June is continuously increasing, favoring the appearance of changes in the local microclimate.

Although it is generally known that the biology of Asian carp kept in fish farms is directly tied to their artificial reproduction, little is known about how they behave in the Danube River. These fish species require a certain environmental requirement to be met in their endemic habitats in the Asian rivers in order to achieve normal natural reproduction. The following features of these conditions are the most crucial between 18 °C and at least 22 °C, the water’s temperature stabilized for a comparatively long amount of time amd the persistence of the rising water level and its existence was up to 3 m^3^/second of water flow. Additionally necessary is a specific amount of turbidity in the water and the proper locations for the development and upbringing of the progeny in the Danube River that the condition of silver carp spawning stimulation is also met [16].

As a result, there is not much of a reason for Asian carp to avoid spawning in the Danube River’s main channel given the environmental factors. This is a sign that the fish population in this area is distributed quite uniformly, or, more likely, that adult silver carp are actually migrating upstream for reproduction from the lower reaches of the river, where they are looking for suitable spawning grounds [57]. The silver carp migration typically takes place in June. Even in years like 1998, when the adult flocks contained fewer individuals, this feature was evident. The results of an experimental fishing operation in various locations upstream of the Danube Delta, where only individuals weighing more than 4 kg possible spawners were recorded, provided conclusive proof of the massive presence of Asian carp throughout the migration period [57].

The possibility for natural reproduction of *Hypophthalmichthys molitrix* and *Hypophthalmichthys nobilis* has received little attention despite being widely introduced throughout Europe, the majority of the time accidentally. We looked into the spawning environments and the presence of young-of-the-year *Hypophthalmichthys nobilis* in an irrigation canal (in Danube Delta polder area) network between 2011 and 2015. The canal network’s adult bighead carp population consisted of big, probably mature fish with an average density of 45.2 kg/ha (over 10-fold more than in the main river). The 29 young bighead carp discovered ranged in size from 7.4 to 13.1 cm (TL) and 9.5 to 12.7 g. These juveniles were estimated to be 94–100 days old based on otolith-derived spawning dates, which put their fertilization and hatch dates around the mid-to-end of June [58]. The species is more prevalent upstream from the delta due to its rheophilic nature and young specimens of *Hypophthalmichthys molitrix* were observed in great numbers in the lakes of the upper Danube Delta in 1992 [59].

As a result of effective natural spawning, which depends on two factors: water temperatures over 22 °C and increased river velocity following spring and summer rainfalls, from 2 to 5 km/h; the Danube River can offer optimal conditions for a significant number of offspring and brood in some years [59]. Additionally, through our own expeditions in the Danube Delta in the summer of 2020, in August we observed a shoal of 1-year-old fry of Asian silver carp belonging to the species *Hypophthalmichthys molitrix*, on the Sontea channel in the Danube Delta. Thus, without any doubt, we can say that the species reproduces naturally at least in the Danube Delta, as well as in the lower Danube.

## 5. Nature of Nutrition

The silver carp has a unique feeding behavior, which is distinguished from the usual movement by rapidity and vigor. The fish moves quickly when eating. Fish quickly absorb water, seal their mouths, and discharge it through the operculum when feeding. Silver carp fry appear to be active filter feeders [60]. They do not feed at very low particle densities [61]. Silver carp feeding spectrum includes a wide variety of species of phytoplankton and algae that can be found in the Danube River. The most prevalent species are diatoms and green algae. Due to the high turbidity of the water, cyanobacteria, and euglena in the Danube are not highly diverse and do not achieve huge numbers. Comparatively, some zooplankton can be found in the Danube channel but the average number of organisms, mostly rotifers, and copepods, does not exceed 7000 in. (individuals)/m^3^ during the vegetation period and increase to 14,000 in. m^3^ in the summer [62].

## 6. Ecological Effect and Conclusions

Since little is known about silver carp as an invader species in Europe, it is challenging to predict the future ecological threats it involves. The majority of Asian carp prediction models have prioritized the probable spread regions over the potential ecological implications [63,64,65].

The environmental changes related to the damming of the major part of the Danube River contributed to the decreasing number of fish species and to a general decrease in population size. In contrast, due to its extremely high fecundity (up to 5 million eggs/mature female), the silver carp can quickly take over and outcompete native species causing enormous damage to the entire ecosystem. There is abundant literature showing that their numbers alone will quickly alter trophic niches of the native fish communities. This is also due to the fact that silver carp individuals consume up 40% of their body weight in phytoplankton and zooplankton every day; therefore, they can quickly reduce the food availability for juvenile fish, forage fish, native filter-feeding fish (shad), mussels, insects, and other organisms that completely rely on phytoplankton and zooplankton as their only food source [64,65]. Taking into account the information obtained from the Romanian scientific literature, as well as the international literature dedicated to this species, we can draw some conclusions regarding the existence of the species in the Danube Delta, as well as in other suitable habitats.

Furthermore, *Hypophthalmichthys molitrix* has been reported, as previously mentioned, in a large variety of water bodies on Romanian territory, some larger than 5 ha (hectares) but not in lakes located in the mountain area, where water temperature is too low during summer times for the species to reproduce.

In conclusion, the phenomenon of increasing temperatures for May–June, leads to changes in the local microclimate, which in turn can have serious implications in the evolution of the ichthyofauna. This is facilitating the expansion of some species, such as the highly prolific silver carp, at the expense of others which are more sensitive to rapid climate and environmental changes.

Therefore, it can be certainly said that the species investigated reproduces naturally in Romanian rivers where the water temperature reaches or exceeds 22 °C at the beginning of June, meaning in the lower course of the Romanian Danube, its tributaries, and the Danube Delta.

## Figures and Tables

**Figure 1 life-12-01582-f001:**
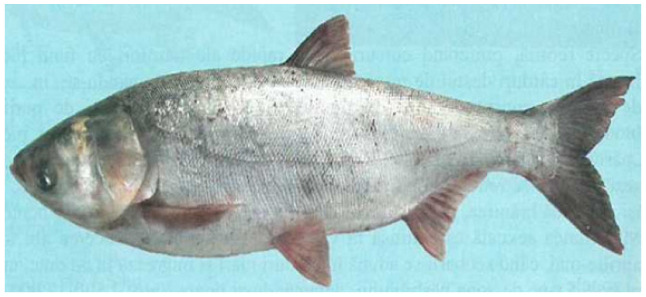
*Hypophthalmichthys molitrix*, image reproduced with permission from Ref. [24]. Copyright year 2007, copyright owner’s Vasile Otel [22]. Length: 40 cm (exceptional over 1.5 m). Weight: 1–2 kg (exceptionally over 60 kg). Appearance: Very large head with low-set eyes, large mouth, strong pharyngeal teeth. The body is covered with small silvery scales [22].

**Figure 2 life-12-01582-f002:**
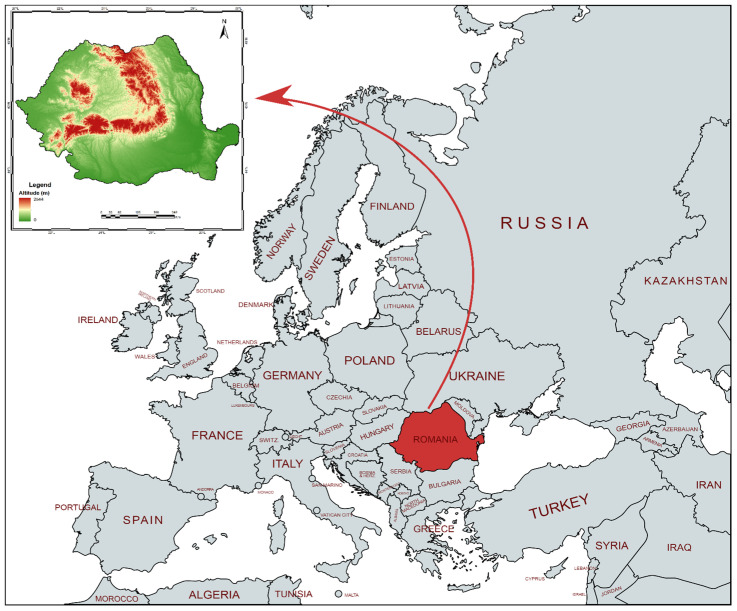
The political map of Europe with the location of Romania in the lower basin of the Danube.

**Figure 3 life-12-01582-f003:**
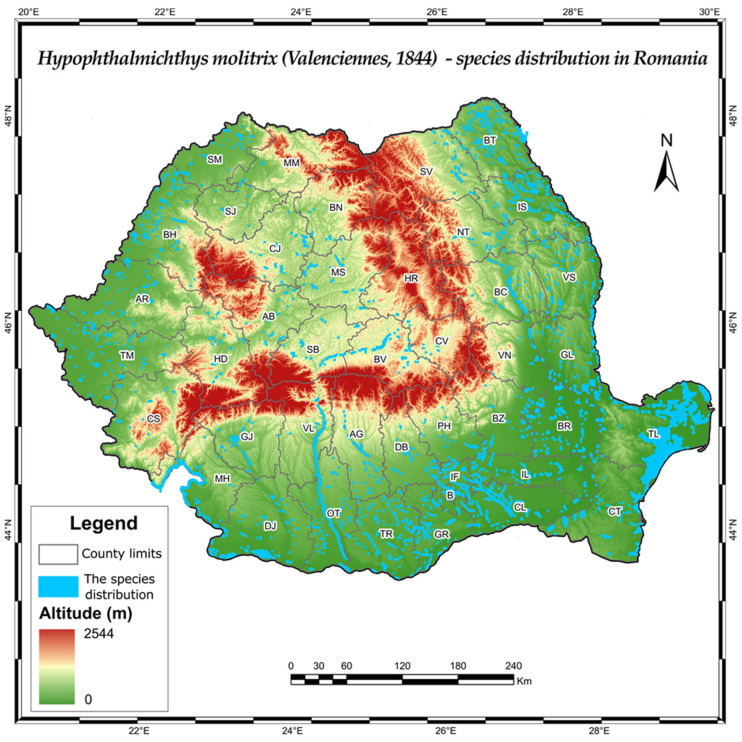
The distribution of the species *Hypophthalmichthys molitrix*, in the actual territory of Romania. Original map made based on data from the cited literature in this review material.

**Figure 4 life-12-01582-f004:**
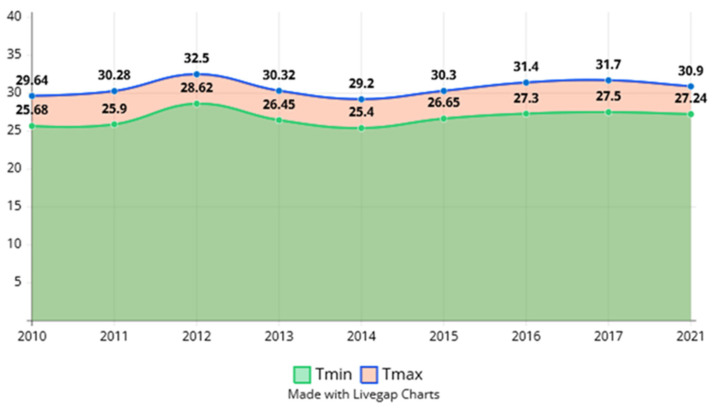
The graph above show the general trend of temperature increase for May and the beginning of June, a trend accentuated by the phenomenon of global warming and prolonged sects at the local level along the Danube.

**Figure 5 life-12-01582-f005:**
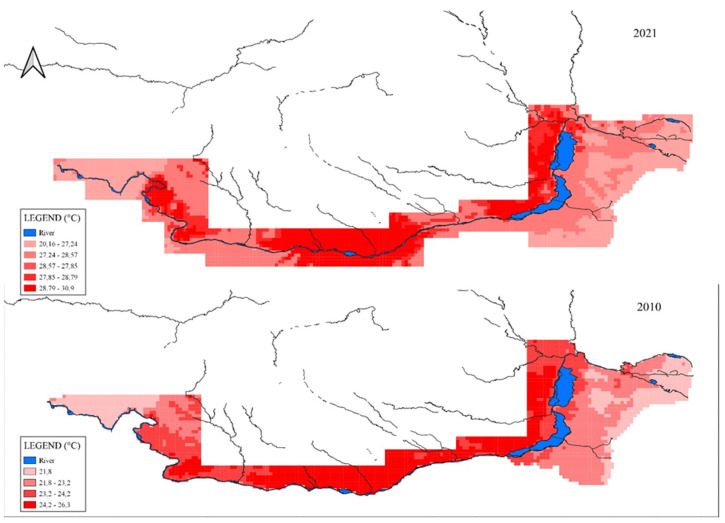
Danube water temperatures in Romania in May–June 2010 below, 2021 above. Original map based on CRU weather data combined with local station data.

## Data Availability

Not applicable.

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
