# Peer review of "Silver Carp (Hypophthalmichthys molitrix) (Asian Silver Carp) Presence in Danube Delta and Romania—A Review with Data on Natural Reproduction"

_life, 2022, doi:10.3390/life12101582_

Round 1

Reviewer 2 Report

This paper provides new information about the breeding of fallen silver carp in the natural environment of the Danube delta.

Author Response

Thank you for your consideration. The manuscript has been improved taking into account all review reports.

Reviewer 3 Report

This is an interesting review. I provide some suggestions for making it more readable and clear.

Somewhere in the first section you should provide a clear and concise objectives statement. The introduction has most of the information needed, but it does not create a clear roadmap for the rest of the paper. Please organize your arguments more carefully in the introduction to show a need for the current review, and provide a clear objective statement to guide readers through the paper.

Second, the manuscript needs better paragraph level organization (the flow of ideas from one paragraph to another) and better organization within paragraphs. Many paragraphs have multiple ideas jumbled together as a series of apparently disconnected statements. Please provide a key sentence for each paragraph and make sure each paragraph treats only one idea.

In some places there is redundant information given. For example, lines 120 and 121 are redundant with the previous text. There are several places where information is difficult to understand because it is not clear how it contributes. 

Round 2

Reviewer 1 Report

The current reviewed MS has been improved by authors. Only several details should be considered as follows:

1.     The title may be changed to “ Silver carp (Hypophthalmichthys molitrix) presence in Danube Delta and Romania – a review with data on natural reproduction”

2.     Line 56, “these are” should be changed to “they are”

3.     Line 66, “Asian carps of the genus Hypophthalmichthys spp.” should be changed to “Asian carps of the genus Hypophthalmichthys

4.     Line 75, “known as silver carp respectively bigheaded carps” should be changed to “known as silver carp and bigheaded carps respectively”.

5.     Line 182, deleted comma after “natural waters”.

6.     Line 296, changed “Asian silver carp” to “ silver carp”.

Reviewer 3 Report

Needs some additional minor revisions, especially in English language usage. 

Line 60 As noted in the previous paragraph, Asina carp are not a single species, but several. I suggest rewording this sentence as follows: “Asian carp are widely cultured and traded all over the world due to their fast growth rate…”

Line 75 “…known as silver carp and bigheaded carp, respectively, play…”

Line 84, delete “and is” at end of sentence.

Line 98, use “reproduction” not reproducing.

Line 148-149, do not use “he” rather use “it”.

Line 152 – sentence does not make sense, use fecundity (not prolificacy) and change the structure of the sentence.

Line 163 – rewrite sentence, doesn’t make sense.

Lines 187-190, Redundant material, delete.

Line 357 the phrase “The effect is one in evolution,” makes no sense. Delete and change the sentence to clarify.

Lines 404-407, Avoid one sentence paragraphs. (See lines 455 to 458 and other palces as well)

Line 428, once again you refer to Asian carp as a species. Fix this sentence to be consistent with previous text.
